# Genome-Scale Characterization of *Mycobacterium abscessus* Complex Isolates from Portugal

**DOI:** 10.3390/ijms242015402

**Published:** 2023-10-20

**Authors:** Sofia Carneiro, Miguel Pinto, Sónia Silva, Andrea Santos, Irene Rodrigues, Daniela Santos, Sílvia Duarte, Luís Vieira, João Paulo Gomes, Rita Macedo

**Affiliations:** 1National Reference Laboratory for Mycobacteria, Department of Infectious Diseases, National Institute of Health Doutor Ricardo Jorge (INSA), 1649-016 Lisbon, Portugal; sofia.carneiro@insa.min-saude.pt (S.C.); andrea.santos@insa.min-saude.pt (A.S.);; 2Department of Life Science, NOVA School of Science and Technology, NOVA University Lisbon, 2829-516 Lisbon, Portugal; 3Genomics and Bioinformatics Unit, Department of Infectious Diseases, National Institute of Health Doutor Ricardo Jorge (INSA), 1649-016 Lisbon, Portugal; miguel.pinto@insa.min-saude.pt (M.P.); j.paulo.gomes@insa.min-saude.pt (J.P.G.); 4Technology and Innovation Unit, Department of Human Genetics, National Institute of Health Doutor Ricardo Jorge (INSA), 1649-016 Lisbon, Portugal; daniela.santos@insa.min-saude.pt (D.S.); silvia.duarte@insa.min-saude.pt (S.D.);; 5Veterinary and Animal Research Centre (CECAV), Faculty of Veterinary Medicine, Lusófona University, 376 Campo Grande, 1749-024 Lisbon, Portugal

**Keywords:** *Mycobacterium abscessus*, MABC, nontuberculous mycobacteria, whole-genome sequencing, wgMLST, antimicrobial resistance, genomic diversity, Portugal

## Abstract

The *Mycobacterium abscessus* complex (MABC) is an emerging, difficult to treat, multidrug-resistant nontuberculous mycobacteria responsible for a wide spectrum of infections and associated with an increasing number of cases worldwide. Dominant circulating clones (DCCs) of MABC have been genetically identified as groups of strains associated with higher prevalence, higher levels of antimicrobial resistance, and worse clinical outcomes. To date, little is known about the genomic characteristics of MABC species circulating in Portugal. Here, we examined the genetic diversity and antimicrobial resistance profiles of 30 MABC strains isolated between 2014 and 2022 in Portugal. The genetic diversity of circulating MABC strains was assessed through a gene-by-gene approach (wgMLST), allowing their subspecies differentiation and the classification of isolates into DCCs. Antimicrobial resistance profiles were defined using phenotypic, molecular, and genomic approaches. The majority of isolates were resistant to at least two antimicrobials, although a poor correlation between phenotype and genotype data was observed. Portuguese genomes were highly diverse, and data suggest the existence of MABC lineages with potential international circulation or cross-border transmission. This study highlights the genetic diversity and antimicrobial resistance profile of circulating MABC isolates in Portugal while representing the first step towards the implementation of a genomic-based surveillance system for MABC at the Portuguese NIH.

## 1. Introduction

Species of the *Mycobacterium abscessus* complex (MABC; which include subspecies *abscessus*, subsp. *massiliense*, and subsp. *bolletii*) are the most reported rapidly growing nontuberculous mycobacteria (NTM) responsible for severe respiratory, skin, and mucosal infections [1,2,3]. As NTM are environmental organisms, MABC infections are assumed to occur after exposure to a contaminated environment. Although infections are more likely to occur indirectly via fomite contamination or long-living infection aerosols, person-to-person transmission in cystic fibrosis (CF) patients may also occur [4,5,6,7,8]. For example, a genomic study focused on isolates from hospitalized patients at a CF center showed that 31.5% of patients were infected with the same (or a very similar) strain of MABC, suggesting nosocomial transmission [6]. The same observations regarding possible person-to-person transmission were also demonstrated in another study conducted in a CF center in Seattle [9]. Given the increase in MABC cases reported worldwide [10,11], in 2016, Bryant and colleagues demonstrated the usefulness of whole-genome sequencing (WGS) in the study of MABC isolate transmission models, revealing the existence of dominating circulating clones (DCCs), defined by clusters of genetically related isolates of *Mycobacterium massiliense* or *Mycobacterium abscessus* [4]. These have been associated with the majority of human infections, higher antibiotic resistance patterns, and worse clinical outcomes [4,6]. In addition, these isolates have genomic characteristics that allowed this NTM to evolve from an environmental organism to a true pulmonary pathogen [4,6].

Additionally, MABC species are emerging multidrug-resistant pathogens with particular clinical relevance, causing lung disease and mainly affecting immunocompromised people [3]. Due to their innate and acquired resistance to a wide range of antimicrobials, and the limited therapeutic options [12], MABC infections are challenging and sometimes impossible to treat [13]. Several genetic targets have already been associated with resistance [14], such as mutations in the ribosomal RNA genes *rrs* (16S rRNA) and *rrl* (23S rRNA) resulting in resistance to amikacin and macrolides, respectively [15]. In *M. abscessus* and *Mycobacterium bolletii*, the *erm(41)* gene is responsible for an inducible innate resistance to macrolides, whilst isolates of *M. massiliense* have this gene inactivated and can therefore be susceptible [16]. This emphasizes the need to correctly identify the MABC subspecies in order to predict resistance phenotypes and possible treatment outcomes [17].

Currently, there are no defined guidelines for standard therapy, and treatment regimens are based on a multidrug approach with long-term administration, which can lead to toxicity and the emergence of new resistance phenotypes [18,19,20,21,22]. For this reason, drug susceptibility testing (DST), following the Clinical Laboratory Standards Institute Guidelines (CLSI), should always be performed to guide treatment regimen design [23,24]. However, given the poor correlation between in vitro data and clinical efficacy results [25], i.e., laboratory issues that lead to low accuracy in phenotypic prediction, WGS-based approaches may be key to the prediction of resistance through the screening of multiple resistance-associated genetic markers and, consequently, supporting treatment regimen definitions, as has been demonstrated for other bacterial pathogens [26,27,28].

In this context, in the present study, we characterized the genetic diversity and antimicrobial resistance profiles (using molecular, genomic, and phenotypic approaches) of 30 MABC isolates (collected between 2014 and 2022) from the National Reference Laboratory for Mycobacteria (NRL-TB) of the Portuguese National Institute of Health (NIH). For this purpose, we applied a gene-by-gene approach (wgMLST) in order to understand the genetic similarities between Portuguese strains and those identified in other countries. Additionally, we performed DST for seven antimicrobials and screened isolates for mutations that are possibly associated with antimicrobial resistance using molecular and genomic approaches.

## 2. Results

### 2.1. Portuguese Dataset Characterization

The 30 MABC isolates enrolled in the present study were retrieved from 23 patients (12 males, 10 females, 1 of unknown sex) with a median age of 69 ± 16 (Table 1). Regarding region of residence, 13 patients were from the Lisbon Metropolitan Area (LMA), 3 were from the Algarve, 1 was from the North Region, and 1 from the Central Region. For five patients, the region of residence was unknown. Molecular methods allowed the identification of ten isolates as *M. abscessus*, six as *M. massiliense*, ten as *M. bolletii*, and four as MABC species (no differentiation was possible). WGS analysis allowed for the differentiation of all 30 isolates at the subspecies level: 16 isolates as *M. abscessus*, 6 as *M. massiliense*, and 8 as *M. bolletii* (Table 1). Throughout this manuscript, we will use the identification results provided by the WGS approach.

### 2.2. Antimicrobial Susceptibility Profiling

The molecular and phenotypic antimicrobial resistance (mDST and pDST) profiles of all Portuguese (PT) isolates are presented in Figure 1 (the MICs of pDST are shown in Appendix A). According to the mDST results, 29 out of the 30 isolates carried the wild-type *erm(41)*T28 genetic marker, which is known to be responsible for induced resistance to macrolides. Of note, although all PT *M. massiliense* strains carry the wild-type *erm(41)*T28, these subspecies are known to have a deletion in *erm*, which can only be detected with gDST (see the Section 4), rendering it inactive, which can result in a macrolide-susceptible phenotype. One *M. abscessus* isolate (PTNTM_0027) did not present the wild-type *erm(41)*T28, in agreement with the pDST results. No resistance-associated mutations in the *rrs* gene were detected through mDST, suggesting potential susceptibility to aminoglycosides.

pDST performed on twenty-seven isolates showed that most isolates were susceptible to amikacin (AMK) (Figure 1), with only three isolates showing an intermediate phenotype (two *M. bolletii* and one *M. massiliense*). In contrast, the majority of isolates were resistant to doxicicline (DOX) (77.78%; 21/27), fluoroquinolones (85.19%; 23/27), and claritromycin (CLR; 74.07%; 20/27). Although CLR is one of the current antimicrobial drugs of choice for treatment, all *M. bolletii* isolates, 14 out of 16 *M. abscessus* and only 1 *M. massiliense* (PTNTM_0030), were resistant (Figure 1). The majority of the isolates (88.89%; 24/27) were resistant to more than two antimicrobials. Noteworthily, one isolate was resistant to all antimicrobials tested (PTNTM_0029) and one isolate (PTNTM_0022) was fully susceptible (Appendix A).

WGS data were used to screen a total of 48 resistance-associated genetic markers (gDST) for different classes of antimicrobials (Figure 1; a detailed list of markers is presented in Appendix A). Regarding aminoglycosides, the resistance-associated mutations A1374N and A1375N in the *rrs* gene (MAB_r5051) were not observed in any isolate. On the other hand, all isolates carried the aminoglycoside phosphotransferases MAB_3637c, MAB_4910c, MAB_0951, and MAB_0327, which are believed to be linked to resistance in *M. smegmatis* and hypothesized to have the same function in MABC [12]. Only one isolate (PTNTM_0011) presented a deletion in MAB_2385 (encoding a Poly(A) in 3″-O-phosphotransferase), suggesting streptomycin susceptibility [29]. All isolates carried the MAB_4395 gene, coding for a 2′-N-acetyltransferase, which has been described to be associated with a higher MIC for gentamicin, tobramycin, and kanamycin [30]. Additionally, all our isolates also carried the MAB_4532c gene (*eis2*) described, in vitro, as being responsible for impairing hygromycin and AMK activity through structural modifications (acetylation of the primary amine of aminoglycosides) [31,32].

The ethambutol-resistance-associated mutations I303Q and L304M in the *embB* gene (MAB_0185c) [33] were detected in all isolates. Potential reduced susceptibility to β-lactams (i.e., imipenem and cefoxitin) was observed in all isolates, due to the presence of beta-lactamase Bla_Mab_ (MAB_2875). Resistance to thioacetazone can be conferred by the presence of a complex efflux pump, MmpS5-MmpL5 (MAB_4383c/MAB_4382c) [34,35]. Most MABC isolates had this efflux pump, except for all *M. massiliense* isolates and two *M. abscessus* isolates (PTNTM_0006 and PTNTM_0027), suggesting a potential increased susceptibility to thioacetazone for the latter (Appendix A). Although 85% of isolates were phenotypically resistant to ciprofloxacin (CIP) and/or moxifloxacin (MXF) (Figure 1), no fluoroquinolone-resistance-associated mutations in *gyrA* and *gyrB* were detected (Appendix A).

Regarding macrolide resistance, only one isolate (PTNTM_0027) showed an *erm(41)*C28 genotype associated with macrolide susceptibility, which is concordant with the mDST and pDST results. *M. massiliense erm(41)* deletion leading to gene inactivation was confirmed in all isolates via WGS. The isolate PTNTM_0008 also showed the macrolide-resistance-associated mutation A2270N in the *rrl* gene [36] (Figure 1). Other resistance markers in the *rrl* gene (G795A, A2271N, G2281N, or A2293N) were screened for, but none were found in any of our isolates (Appendix A). PT isolates were additionally screened for recently proposed novel genetic markers potentially associated with CLR resistance [19]. All our isolates presented at least one mutation in one of these genes (Figure 1 and Appendix A), suggesting their potential impact on CLR decreased susceptibility.

Of note, in some cases, pDST was not even concordant among same-patient isolates. In particular, isolates from patient 3 showed differences in phenotypic susceptibility for FOX, CLR, DOX, and LNZ, and isolates from patient 4 for AMK and MXF, while presenting the same analyzed resistance-associated genetic markers.

### 2.3. MABC Phylogenetic Analysis

Reconstruction of the global core-genome Multi-Locus Sequence Typing (cgMLST)-based phylogeny presented by Diricks and colleagues [37] (Figure 2) allowed us not only to assess the 250 allelic differences (ADs) cut-off for DCC classification but also to classify our own isolates. In fact, clustering analysis at all AD thresholds on this dataset revealed a large cluster stability interval ranging from 95 to 297 ADs in the MST, where the proposed DDC classification thresholds fall. Noteworthily, an earlier stability region was also observed ranging from 56 to 93 ADs, which could potentially be used for long-term genomic surveillance of circulating MABC strains. After successfully reproducing the schema, we were able to classify eleven Portuguese MABC isolates into different DCCs (Figure 2 and Table 1). As such, the isolates PTNTM_0014, PTNTM_0022, PTNTM_0023, PTNTM_0024, PTNTM_0028, and PTNTM_0029 were classified into DCC1, the isolates PTNTM_0010, PTNTM_0021, and PTNTM_0030 were classified into DCC3a, the isolate PTNTM_0008 was classified into DCC4, and the isolate PTNTM_0012 was classified into DCC6.

To take advantage of the adapted wgMLST schema, we applied a dynamic approach on our PT genomic dataset to allow us to maximize the number of shared loci, thus increasing the sensitivity of the analysis. The global genetic diversity of the PT isolates is presented in Figure 3, which clearly shows that most isolates are highly divergent, suggesting the circulation of a wide range of MABC strains in Portugal. In fact, eleven isolates presented more than 100 ADs from their closest isolate, with most differing in more than 2000 ADs. Moreover, isolates from the same patient were closely linked (maximum of 16 ADs) in four distinct genetic clusters (Figure 3), while also being genetically distant from other isolates (minimum of 43 ADs observed between PTNTM_0014 and PTNMT_0024). Of note, the genetic cluster composed of isolates PTNTM_0001 up to PTNTM_0005 corresponds to an in-depth studied persistent infection by *M. bolletii* observed in a single patient from 2015 to 2021 [38].

Taking advantage of the large number of available genomic data, we performed an in-depth clustering analysis of the PT isolates by highlighting the 15 closest MABC isolates within the global dataset (Figure 4). Using a conservative approach, i.e., considering isolates linked at <50 ADs within each zoomed-in cluster (close to the identified stability region; see above), twelve PT isolates presented more than 50 ADs to the closest international genome (Figure 4A,C,E,G,L), suggesting that these strains are yet to be reported in other countries. With the exception of PTNTM_0022 and PTNTM_0027 (Figure 4K,N), all remaining PT isolates (ten isolates) integrate multi-country genetic clusters composed of isolates from at least three countries. Curiously, for six PT isolates, the closest genome in the dataset was collected from the United Kingdom (Figure 4B,D,I–K,N). Of note, ten PT isolates (which included all same-patient isolates, PTNTM_0001 up to PTNTM_0005, from Figure 4A) presented more than 2300 ADs from the closest isolate in the analyzed dataset (PTNTM_0006, PTNTM_0007, PTNTM_0009, PTNTM_0011, and PTNTM_0018).

**Figure 2 ijms-24-15402-f002:**
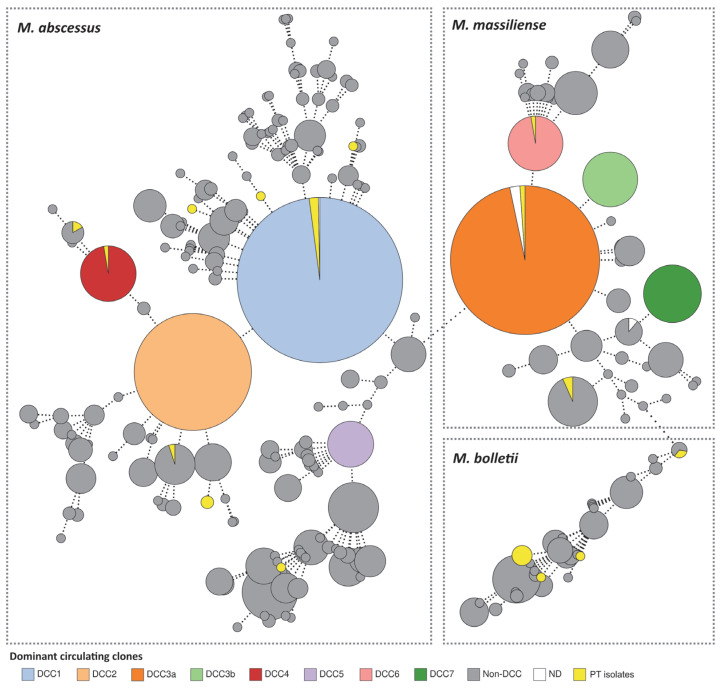
Phylogenetic integration of the Portuguese isolates into the MABC dataset described by Diricks et al., 2022 [37]. Minimum spanning tree was constructed using Grapetree [39] based on the allelic matrix of 1815 isolates and 2648 loci generated with the MABC cgMLST schema, with a site inclusion tolerance of 1% missing data. The size of each node proportionately corresponds to an isolate/genome or group of isolates, and nodes are colored according to the DCC classification. PT isolates are highlighted in yellow. For visualization purposes, nodes have been collapsed at the DCC defining the 250 AD threshold described by Diricks et al., 2022 [37]. Dashed lines connecting nodes represent a number of ADs above 100. ND—not determined.

**Figure 3 ijms-24-15402-f003:**
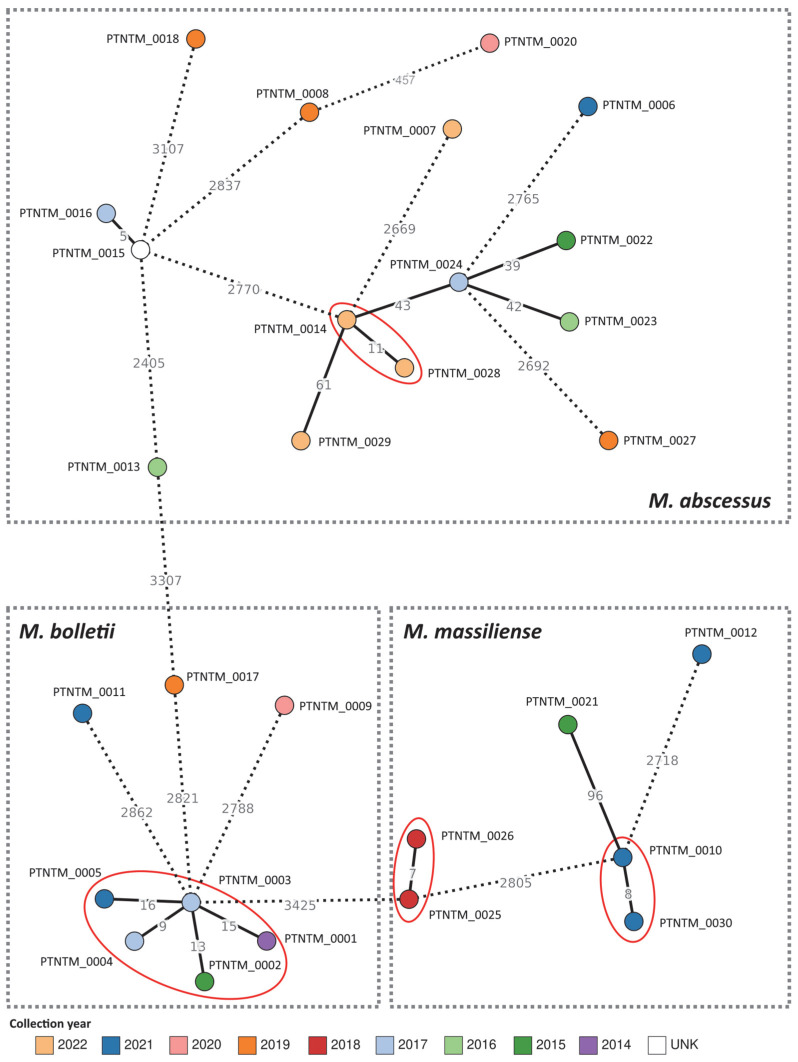
Global phylogeny of the MABC Portuguese isolates based on a gene-by-gene approach. Minimum spanning tree was constructed using Grapetree [39] based on the allelic matrix of 29 isolates and 3458 loci generated with the MABC wgMLST schema, with a site inclusion tolerance of 1% missing data. Each node corresponds to an isolate/genome and has been colored according to its diagnosis year. Isolates belonging to the same patient are highlighted with red circles. Dashed lines connecting nodes represent a number of ADs above 100. UNK—unknown.

**Figure 4 ijms-24-15402-f004:**
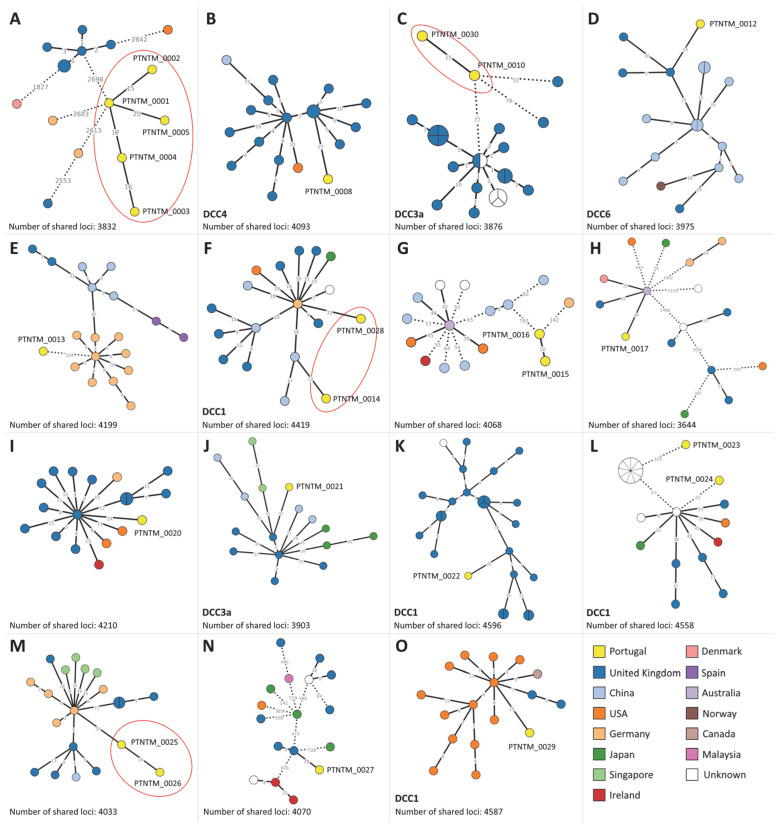
Genetic clusters including PT isolates and the 15 closest isolates of the used public dataset of 7078 genomes. (**A**–**O**) correspond to different genetic clusters enrolling PT isolates. Minimum spanning trees were constructed using Grapetree [39] based on the allelic matrices generated with the MABC wgMLST schema, with a site inclusion tolerance of 1% missing data. At the bottom of each panel, the number of loci under analysis for each subset, i.e., shared by all isolates, is indicated. Each node corresponds to an isolate/genome and has been colored according to its country of origin. Isolates belonging to the same patient are highlighted with red circles. Dashed lines connecting nodes represent a number of ADs above 50.

## 3. Discussion

Infections caused by MABC species are increasing worldwide [3] and are particularly challenging to treat, mainly due to resistance to several commonly used antimicrobials, leading to multidrug regimen treatments and often to treatment failure [40,41]. In the present study, we used molecular, phenotypic, and genomic approaches to characterize MABC isolates circulating in Portugal between 2014 and 2022. WGS allowed for the subspecies classification of all isolates, showing a higher discriminatory power when compared with the molecular methodologies commonly used in the laboratory for routine diagnostics purposes (namely, the *hsp65* sequencing and genotype kits). Among the MABC subspecies, *M. abscessus* was predominantly detected within the PT dataset (16/30) (Table 1), which is consistent with previous studies [42,43].

With regard to the antimicrobial susceptibility tests, pDST confirmed the previously described high levels of resistance associated with the MABC isolates [12]. The antibiotics that were less effective against our isolates were DOX, CIP, and CLR, with resistance rates of 77.78%, 74.07% and 74.07% (Figure 1 and Appendix A), respectively, in line with previous observations. For example, a study performed in the United Kingdom considering 58 MABC isolates showed high resistance rates of 98% for DOX and 95% for CIP [44], and another performed in China using 20 MABC isolates of all three subspecies showed 70% and 95% resistance rates to DOX and CIP, respectively [45]. Most of the PT isolates that showed phenotypic resistance to CLR were *M. abscessus* and *M. bolletii*. Concerning *M. abscessus*, these high resistance rates (i.e., more than 80% CLR resistance) were also reported in studies from China [46] and Japan [47]. In contrast, a larger study evaluating 383 MABC clinical isolates, also from China, between 2014 and 2016 revealed much lower levels of CLR resistance (only 6.4% and 10.4% for *M. abscessus* and *M. massiliense*, respectively) [48]. As expected, and also supported by previous studies [44,46,49], AMK seems to be effective for MABC species as 88.89% of the PT isolates were susceptible to this aminoglycoside (Figure 1), which is currently used for first-line treatment. However, it is clearly recognized that phenotypic in vitro testing has a poor correlation with the in vivo activity of antimicrobials [50]. In fact, our results seem to support this phenomenon, as, for example, for patient 3 (PTNTM_0010 and PTNTM_0030) for CLR and patient 4 (PTNTM_0014 and PTNTM_0028) for MXF, pDST showed acquisition of resistance, whilst the gDST results were identical in both isolates (Appendix A). This observation supports either the lack of sensitivity of pDST or the presence of unknown genetic markers with an impact on resistance. Nevertheless, the use of WGS for the screening of specific resistance-associated genetic markers has the potential to be a powerful alternative approach for treatment regimen recommendation. In fact, in the present study, by using WGS, it was possible to analyze a wider panel of antimicrobials that could not otherwise be tested in vitro or via mDST (Appendix A). Although our findings indicate low levels of phenotypic resistance to AMK, all isolates presented aminoglycoside phosphotransferases (MAB_3637c, MAB_4910c, MAB_0951, and MAB_0327) and the *eis2* gene, all of which have been previously associated with resistance to this class of antimicrobial [12,32]. This highlights the need for additional large-scale phenotype/genotype studies either to identify novel resistance-associated markers or to understand the incongruence between resistance profiles, as the presence of the screened molecular markers may still be responsible for low-level resistance. Consequently, when using solely pDST, MABC isolates are likely to be classified as having a susceptible or intermediate resistance phenotype, prolonging potential inadequate treatment regimens that could be leading to the establishment of fully resistant strains. Furthermore, continued exposure to AMK has already been hypothesized to be a cause for the emergence of AMK resistance, highlighting the importance of the early detection of resistance markers [51].

As mentioned before, the successful macrolide-based treatment regimens for *M. massiliense* infections are related to its intrinsic susceptibility associated with an inactive *erm(41)* gene. This inactivation is due to a large deletion [17] that can only be detected through genomic sequencing, while mDST will only detect inactivation by a truncated C-terminal region (T28C) (Appendix A). In fact, all of our *M. massiliense* isolates showed this deletion, and just one (PTNTM_0030) exhibited a phenotypic resistance to CLR, probably due to another resistance mechanism that we could not predict. Due to the high CLR resistance observed in *M. bolletii* and *M. abscessus* species, we suggest that its use as a first-line treatment option may require revision. The introduction of intravenous β-lactams, such as imipenem or FOX combined with AMK, was one of the approaches [52,53]. Here, only two isolates were shown to be phenotypically resistant to FOX, one *M. massiliense* (PTNTM_0012) and one *M. abscessus* (PTNTM_0016) (Figure 1). Resistance to β-lactams was previously reported as being related to the presence of an endogenous β-lactamase (Bla_mab_) encoded by MAB_2875 that reduces the activity of FOX and imipenem [54,55]. WGS showed that all isolates carried a version of this protein, suggesting that pDST may lack sensitivity to FOX. Another example of a potential lack of pDST sensitivity was observed for the isolate PTNTM_0021, which showed only one resistance phenotype (to CIP), while presenting several additional resistance markers to FOX (*Bla_mab_*) [54], ethambutol (*embB*) [33], aminoglycosides (MAB_3637c, MAB_4910c, MAB_0951, and MAB_0327) [12,20], and CLR [19] (Appendix A).

As this is a non-notifiable disease, changes in the public health policy are required in order to monitor MABC infections and better understand its epidemiology. By integrating the Portuguese MABC isolates into the dataset used by Diricks et al. [37], we were able to clearly distinguish MABC subspecies and classify eleven isolates into DCCs (Table 1 and Figure 2). Although DCC isolates have been described to have multidrug resistance, causing infections with worse outcomes, in our dataset, regardless of the DCC, the majority of the isolates were resistant to more than two antimicrobials, suggesting the potential emergence of multidrug-resistant strains. Although we have to take into account the low number of Portuguese MABC isolates included in our study, the applied gene-by-gene analysis (Figure 3) showed that the Portuguese isolates have a high degree of genetic diversity, regardless of the subspecies, with isolates separated by more than 2000 ADs. In fact, several studies have already described high genetic diversity in MABC [4,37,56]. For example, a multi-country study enrolling 1080 clinical MABC isolates revealed extensive genetic differences between isolates both within subspecies and different infected individuals, suggesting disease acquisition from diverse environmental niches [4]. Previous genomic studies have mainly analyzed MABC strains recovered from CF patients due to the high incidence of MABC infections in patients with this disease, showing that these clinical settings are less prone to yielding diversity and supporting human-to-human transmission [6,7,8]. Concordantly, in our dataset, only strains belonging to the same patient clustered at a low AD threshold (below ~20 ADs), suggesting that no human-to-human transmission could be detected (although patient clinical history was not available). Despite the fact that isolates belonging to DCCs have been associated with patients with CF [4], our results, together with previous studies [57,58,59], demonstrate the presence of these clones in non-CF patients. Furthermore, seven out of eleven Portuguese DCC isolates were clustered with strains isolated in other countries at <50 ADs (Figure 4), suggesting the existence of MABC lineages with potential international circulation or potential cross-border transmission.

Globally, we believe that studies focusing on comparative genomic analyses in non-CF patients are still needed to fully comprehend the distribution and transmission of MABC. Additionally, considering the above-mentioned issues with pDST, gDST will likely be applied to predict resistance, although phenotype/genotype studies in MABC are still required to identify clear resistance-associated markers. Given the degree of genetic diversity observed for MABC circulating in Portugal, WGS-based surveillance of this pathogen may be key both for diagnostic purposes (i.e., subspecies identification and resistance prediction) and for disease monitoring (i.e., current circulation of environmental strains or early detection of possible human-to-human transmission). In this context, we also believe that environmentally sampled MABC isolates should likewise be surveyed from a One Health perspective. As such, this study constitutes the first step in the implementation of a robust surveillance system at the Portuguese NRL-TB.

## 4. Materials and Methods

### 4.1. Sample Collection and Characterization

Between 2014 and 2022, the NRL-TB obtained 56 MABC cultures from 38 patients. Metadata such as age, gender, residence (according to the Nomenclature for Territorial Units level II (NUT II)), and date of diagnosis were collected. The patient dataset consisted of 19 males, 18 females, and 1 of unknown sex, with a median age of 64 ± 18 years. The majority of these patients were from the Lisbon Metropolitan Area (LMA; 34.2%), followed by the North Region (18.4%) and the Algarve (13.2%). Alentejo and the Central Region accounted for two patients each (5.3%), and the Azores Autonomous Region accounted for one patient (2.6%). For 21.7% of the patients (8/38), the region of residence was not available. In total, 30 MABC isolates retrieved from 23 patients were successfully recovered and included in this study for further analysis (Appendix A). More than one MABC isolate from the same patient was included in the analysis when the isolation date was six or more months apart. For molecular and genomic approaches (*hsp*65 gene sequencing, genotype kits, and WGS), total DNA was extracted from solid/liquid cultures using the cetyltrimethylammonium bromide (CTAB) procedure for total nucleic acid extraction, as previously described [60], with slight adjustments: (i) after lysozyme addition, suspensions were incubated overnight at 37 °C; and (ii) after SDS/proteinase K solution was added, suspensions were incubated at 65 °C until complete dissolution.

MABC subspecies were identified using GenoType Mycobacterium CM/NTM-DR (Hain Lifescience, Germany) or *hsp*65 Sanger sequencing, as previously described [61,62,63]. Phenotypic antibiotic susceptibility testing (pDST) was performed on 27 strains (3 isolates could not be recovered for further pDST), according to the Clinical and Laboratory Standards Institute (CLSI), for amikacin (AMK), cefoxitin (FOX), ciprofloxacin (CIP), clarithromycin (CLR), doxycycline (DOX), linezolid (LZD), and moxifloxacin (MXF) [24]. Molecular antibiotic susceptibility testing (mDST) was carried out for all 30 strains using the Genotype *Mycobacterium* NTM-DR Kit (Hain Lifescience, Germany), for macrolide and aminoglycoside resistance prediction, according to the manufacturer’s instructions [62].

### 4.2. Whole-Genome Sequencing, Genome Assembly, and Characterization

Isolates’ total genomic DNA was subjected to Nextera XT library preparation (Illumina, San Diego, CA, USA) before paired-end sequencing (2 × 250 bp or 2 × 150 bp) on either an MiSeq, NextSeq 550, or NextSeq 2000 instrument (Illumina, USA) according to the manufacturer’s instructions (Appendix A). All genome sequences were assembled using the INNUca v4.2.2 pipeline (https://github.com/B-UMMI/INNUca; accessed on 20 September 2023), an integrative bioinformatics pipeline for read quality analysis and de novo genome assembly [64]. Briefly, read quality analysis and improvement were performed using FastQC v0.11.5 (http://www.bioinformatics.babraham.ac.uk/projects/fastqc/; accessed on 20 September 2023) and Trimmomatic v0.38 [65] (with sample-specific read trimming criteria determined automatically based on FastQC report), respectively. Genomes were assembled with SPAdes v3.14.0 [66] and subsequently polished using Pilon v1.23 [67] with Quality Assurance/Quality Control statistics being monitored and reported throughout the analysis. Species confirmation and contamination screening were assessed using Kraken v2 [68] (with the Standard-16 database for 12 September 2022, available at https://benlangmead.github.io/aws-indexes/k2; accessed on 20 September 2023) for both raw reads and final polished assemblies. Additionally, polished genomes were also screened for contamination with the ribosomal Multi-Locus Sequence Typing (rMLST) tool (available at PubMLST; https://pubmlst.org/species-id; accessed on 20 September 2023).

For data integration, all genomes (N = 1797) used for the core-genome MLST (cgMLST) created and validated by Diricks et al., 2022 [37] were retrieved from public databases (when reads were available, de novo assembly was performed). Additionally, read datasets (N = 6145) identified as MABC were retrieved from the European Nucleotide Archive (ENA). Assembly was performed using the INNUca v4.2.2 pipeline, and genomes were excluded when they did not meet at least one of the following criteria: (i) final assembly was not identified as MABC or was flagged as contaminated in Kraken or rMLST species classification; (ii) the final assembly size was <4 Mbp or >6 Mbp; (iii) the maximum number of contigs was >250 in the final assembly; and (iv) samples were identified as duplicates with different ENA run accession numbers. A total of 7065 public genomes were included in downstream analysis (Appendix A).

For antimicrobial-resistance-associated genetic marker screening, quality processed reads were individually mapped against the ATCC 19,977 reference genome sequence (Genbank accession number CU458896.1) using snippy v4.5.1 (https://github.com/tseemann/snippy; accessed on 20 September 2023). All screened genetic markers are described in Appendix A.

### 4.3. Gene-by-Gene Analysis

For gene-by-gene analysis, two panels of loci were retrieved (30 June 2022) from the RIDOM Nomenclature Server (https://www.cgmlst.org/ncs; accessed on 20 September 2023) developed by Diricks and colleagues [37]. These panels include 2904 loci from the “*Mycobacteroides abscessus* cgMLST” schema and an additional 1986 loci from the “*Mycobacteroides abscessus* Accessory” schema (agMLST). In the present study, we adapted both locus panels into a whole-genome MLST (wgMLST) schema for chewBBACA v3.1.0 [69] using the PrepScheme module and a training file generated by Prodigal v2.6.3 from the ATCC 19,977 reference genome (RefSeq accession number GCF_000069185.1). After allele calling with chewBBACA on 7095 genomes (i.e., public dataset plus PT genomes), loci called in less than 10% of the dataset were removed from the schema. This resulted in the removal of 45 and 64 loci from the cgMLST and agMLST schemas, respectively (yielding a final schema size of 2859 for cgMLST and 4781 for wgMLST). Exact and inferred matches were used to construct an allelic profile matrix, where other allelic classifications (see https://github.com/B-UMMI/chewBBACA/wiki; accessed on 20 September 2023) were assumed as “missing” loci. Genomes with less than 95% loci called in the cgMLST schema (i.e., fewer than 2717 loci called) were removed for subsequent phylogenetic inferences [37]. This only occurred for 17 out of the 7095 genomes under analysis (Appendix A), including 1 PT genome (PTNTM_0019). Additionally, we used the 1795-genome dataset described in Diricks et al., 2020 in order to classify our PT genomes into DCCs. Briefly, after allele calling using the cgMLST schema with a tolerance of 1% missing data per locus, we took advantage of the ReporTree v2.0.3 software [70] to determine cluster stability regions within our dataset and compared these with the previously described cut-off for the DCC classification of 250 ADs [37]. PT genomes were then integrated into the dataset and classified based on their phylogenetic placement after minimum spanning tree (MST) generation using Grapetree v1.5.0 with the MSTv2 algorithm [39].

To increase the resolution power of the analysis with the wgMLST schema (which may be key for discriminating cases during epidemiological investigations), we used ReporTree [70], which allows maximizing the shared genome between samples in a dynamic manner, i.e., for each subset of strains under comparison, the maximum number of shared loci between them is automatically used for tree construction. As such, MSTs were generated for the complete PT genomic dataset, and we used the ReporTree’s *zoom-cluster-of-interest* option in order to recalculate the MSTs containing each PT genome and their respective closest 15 public genomes, based on the global allelic distance matrix, to assess their genetic relatedness to other internationally reported isolates.

## Figures and Tables

**Figure 1 ijms-24-15402-f001:**
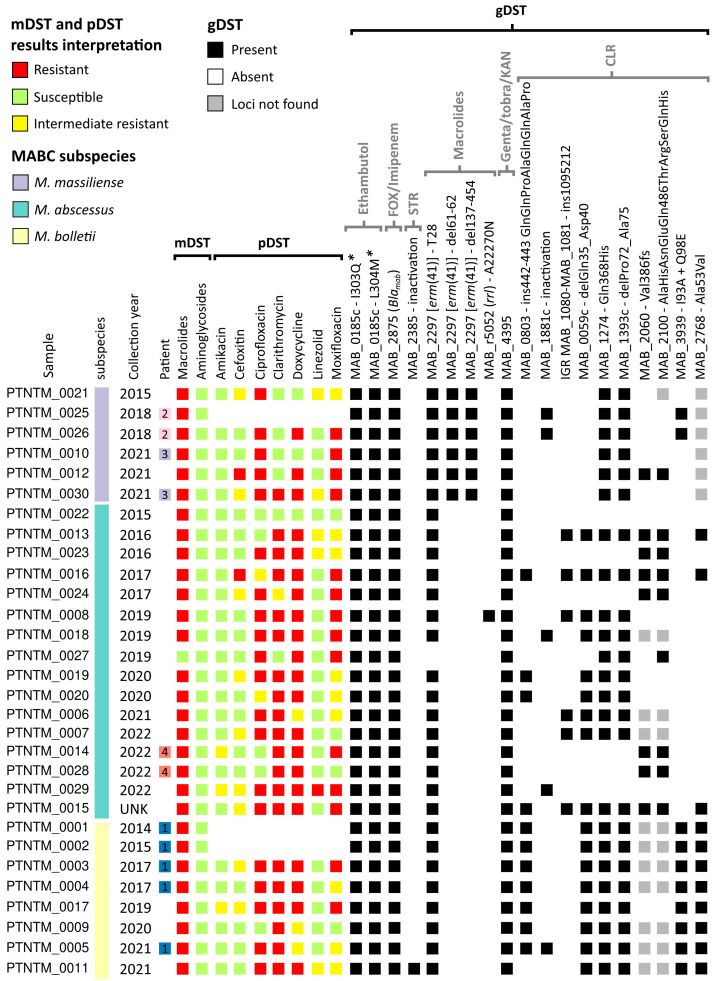
Phenotypic, molecular, and genotypic antimicrobial resistance screening of Portuguese *Mycobacterium abscessus* complex isolates. Gene designations refer to the locus tags of the *M. abscessus* ATCC19977 reference genome. Numbers in the patient column refer to the same patient coding as described in Table 1. * Positions relative to *M. tuberculosis* reference genome H37Rv. mDST—molecular antibiotic susceptibility testing; pDST—phenotypic antibiotic susceptibility testing; gDST—genotypic antibiotic susceptibility testing; FOX—cefoxitin; STR—streptomycin; Genta—gentamicin; tobra—tobramycin; KAN—kanamycin; CLR—clarithromycin.

**Table 1 ijms-24-15402-t001:** Summarized characterization of the Portuguese MABC isolate dataset.

Isolate ID	WGS-Based Taxonomic Classification	DCC Classification	Collection Year	Patient ID	Region of Residence	Gender	Age Group
PTNTM_0001	*M. bolletii*	Non-DCC	2014	Patient 01	Algarve	Male	[65–70[
PTNTM_0002	*M. bolletii*	Non-DCC	2015	Patient 01	Algarve	Male	[65–70[
PTNTM_0003	*M. bolletii*	Non-DCC	2017	Patient 01	Algarve	Male	[65–70[
PTNTM_0004	*M. bolletii*	Non-DCC	2017	Patient 01	Algarve	Male	[65–70[
PTNTM_0005	*M. bolletii*	Non-DCC	2021	Patient 01	Algarve	Male	[70–75[
PTNTM_0006	*M. abscessus*	Non-DCC	2021	Patient 06	LMA	Female	[50–55[
PTNTM_0007	*M. abscessus*	Non-DCC	2022	Patient 17	LMA	Female	[75–80[
PTNTM_0008	*M. abscessus*	DCC4	2019	Patient 11	LMA	Female	[70–75[
PTNTM_0009	*M. bolletii*	Non-DCC	2020	Patient 09	LMA	Female	[65–70[
PTNTM_0010	*M. massiliense*	DCC3a	2021	Patient 03	UNK	Male	[40–45[
PTNTM_0011	*M. bolletii*	Non-DCC	2021	Patient 13	UNK	Male	UNK
PTNTM_0012	*M. massiliense*	DCC6	2021	Patient 14	North	Male	[60–65[
PTNTM_0013	*M. abscessus*	Non-DCC	2016	Patient 10	LMA	Female	[65–70[
PTNTM_0014	*M. abscessus*	DCC1	2022	Patient 04	UNK	Male	UNK
PTNTM_0015	*M. abscessus*	Non-DCC	UNK	UNK	UNK	UNK	UNK
PTNTM_0016	*M. abscessus*	Non-DCC	2017	Patient 21	LMA	Male	[70–75[
PTNTM_0017	*M. bolletii*	Non-DCC	2019	Patient 22	Algarve	Female	[25–30[
PTNTM_0018	*M. abscessus*	Non-DCC	2019	Patient 08	Centre	Female	[80–85[
PTNTM_0019	*M. abscessus*	Non-DCC	2020	Patient 20	UNK	Female	UNK
PTNTM_0020	*M. abscessus*	Non-DCC	2020	Patient 19	LMA	Male	[30–35[
PTNTM_0021	*M. massiliense*	DCC3a	2015	Patient 05	LMA	Male	[75–80[
PTNTM_0022	*M. abscessus*	DCC1	2015	Patient 15	LMA	Male	[75–80[
PTNTM_0023	*M. abscessus*	DCC1	2016	Patient 07	LMA	Male	[55–60[
PTNTM_0024	*M. abscessus*	DCC1	2017	Patient 16	LMA	Female	[65–70[
PTNTM_0025	*M. massiliense*	Non-DCC	2018	Patient 02	LMA	Male	[85–90[
PTNTM_0026	*M. massiliense*	Non-DCC	2018	Patient 02	LMA	Male	[85–90[
PTNTM_0027	*M. abscessus*	Non-DCC	2019	Patient 18	LMA	Female	[75–80[
PTNTM_0028	*M. abscessus*	DCC1	2022	Patient 04	UNK	Male	UNK
PTNTM_0029	*M. abscessus*	DCC1	2022	Patient 12	Algarve	Male	[70–75[
PTNTM_0030	*M. massiliense*	DCC3a	2021	Patient 03	UNK	Male	[40–45[

DCC—dominant circulating clone; LMA—Lisbon Metropolitan Area; UNK—unknown.

## Data Availability

All reads generated for the present study were deposited in the ENA under the study accession number PRJEB57933. Complete details of all isolates and individual ENA run accession numbers are provided in Appendix A.

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
