# Peer review of "Genome-Scale Characterization of Mycobacterium abscessus Complex Isolates from Portugal"

_ijms, 2023, doi:10.3390/ijms242015402_

Round 1
Reviewer 1 Report
I really enjoyed reading and reviewing the article. The authors have done a great job in presenting their work. However, I found several minor mistakes that, if addressed, can improve the manuscript. Please find the attached PDF for a summary of comments.

Author Response
Reviewer #1
I really enjoyed reading and reviewing the article. The authors have done a great job in presenting their work. However, I found several minor mistakes that, if addressed, can improve the manuscript. Please find the attached PDF for a summary of comments.
Line 81: Authors are advised to clarify how they are profiling microbial resistance in MABC at this point.
Response: We have now clarified this sentence.
Line 96: Italicize the species names throughout the manuscript.
Response: Done
Table 1: Please define the subtype of DCC mentioned in the table in the table legend.
Response: Each isolate has its own Dominant Circulating Clone (DCC) coding, as presented in the Table 1. We believe there is no need to specify each coding (referred by the reviewer as subtype).
Line 106: Can the authors explain why hsp65 was not detected in about half of the samples?
Response: In Supplementary Table S1 the abbreviation ND refers to “Not determined” instead of “Not detected”. In fact, it was one of the first analysis performed on our group of samples, but since it proved to have poor discriminatory power for each of the subspecies, it was not performed for all isolates. Still, we included this information in the Table.
Line 107: Please italicize the gene name throughout the manuscript.
Response: Done.
Figure 1: Please use the same annotation as 'T28' from the literature.
Response: Done.
Figure 1: What does the 'Patient' column and the related numbers indicate?
Response: The number refers to the Patient coding as presented in Table 1. We have now added this information to the Figure legend.
Line 128: I believe this is a typo, and the information from lines 128-131 should be included in the Figure 1 legend.
Response: It was indeed a typo. This corresponds to the rest of the Figure’s caption. It is now corrected.
Line 134: Please label the tables in chronological order as they are presented in the manuscript. Table S3 can not come first to Table S2.
Response: We have corrected the order throughout the manuscript.
Line 172: Please elaborate on the abbreviation the first time it is used in the manuscript.
Response: Done.
Line 173: How are different DCCs classified, and what are the criteria for their differentiation?
Response: DCC classifications are based on a previously published study by Diricks et al., (reference 37), that not only developed the cgMLST scheme but also the coding for the different DCC (which are phylogenetically determined).
Line 173: Please elaborate on the abbreviation the first time it is used in the manuscript
Response: Done
Line 176: The criteria for classifying different DCCs are unclear. Please elaborate.
Response: as mentioned in the comment above, DCC were previously proposed by another study. Here, the classification of novel Portuguese isolates was performed using the same rationale, which is based on phylogenetic positioning.
Line 184: Figure 2 also has DCC7?
Response: Indeed, Figure 2 shows the distribution of all DCC within the complete dataset described in Diricks et al., (ref 37), being its purpose to highlight the integration of the Portuguese isolates. No Portuguese isolate belongs to neither DCC7, DCC5, DCC2 nor DCC3b, as stated in the results section.
Line 188: Please explain why PTNTM_0019 was excluded from the analysis?
Response: The exclusion of PTNTM_0019 is described in the methods section (4.3. gene-by-gene analysis; Line 420-422. It was excluded as it had less than 95% loci called in the cgMLST schema.
Reviewer 2 Report
The paper titled "Genome-scale characterization of Mycobacterium abscessus complex isolates from Portugal" by Carneiro and colleagues provides a clear insight into the genome-scale of these strains in Portugal. The authors adequately justify the need for this research and provide substantial methodological support, profiling over 30 strains, which seems essential for the nation's statistical foundation. Proper references are cited, showcasing a solid foundation of knowledge. However, there are some areas where the paper could be improved in terms of detail. It is advisable to address the following comments before resubmitting to IJMS.
1.Abstract:
1-1. If the study is specific to a particular country or region, the keywords should include the name of that country or region.
1-2. The emphasis on data analysis spanning nine years from 2014 to 2022 is somewhat unclear, given the wide range of data usage in the paper. Consider narrowing down the range of years or providing better justification.
2. Introduction:
2-1. The paragraph from line 57 to line 79 is lengthy and somewhat tedious. Split the paragraph and emphasize specific facts in each paragraph to provide a more concise and focused presentation. For example, consider contrasting historical facts from the distant past with more recent facts depending on the timeframe.
3. Results:
3-1. Question: Is there a specific reason why the Age group in Table 1 is primarily limited to the elderly? I'm curious if there was an analysis for the entire age range.
3-2. It might be beneficial to list the data used in Figure 1 by year, unless there is a specific reason for the current arrangement.
4. Discussion:
Since the paper's structure separates Results and Discussion, it's advisable to provide links to specific data discussed in the Discussion section.
5. Consider adding a brief conclusion. This is essential for summarizing the paper effectively.
Minor editing of English language required
Author Response
Reviewer #2
The paper titled "Genome-scale characterization of Mycobacterium abscessus complex isolates from Portugal" by Carneiro and colleagues provides a clear insight into the genome-scale of these strains in Portugal. The authors adequately justify the need for this research and provide substantial methodological support, profiling over 30 strains, which seems essential for the nation's statistical foundation. Proper references are cited, showcasing a solid foundation of knowledge. However, there are some areas where the paper could be improved in terms of detail. It is advisable to address the following comments before resubmitting to IJMS.
1.Abstract:
- If the study is specific to a particular country or region, the keywords should include the name of that country or region.
Response: Done.
1-2. The emphasis on data analysis spanning nine years from 2014 to 2022 is somewhat unclear, given the wide range of data usage in the paper. Consider narrowing down the range of years or providing better justification.
Response: We do not understand the reviewer’s comment. Our study is focused on a dataset of Portuguese isolates collected from 2014 up to 2022.
- Introduction:
2-1. The paragraph from line 57 to line 79 is lengthy and somewhat tedious. Split the paragraph and emphasize specific facts in each paragraph to provide a more concise and focused presentation. For example, consider contrasting historical facts from the distant past with more recent facts depending on the timeframe.
Response: We have now split and simplified the paragraph.
- Results:
3-1. Question: Is there a specific reason why the Age group in Table 1 is primarily limited to the elderly? I'm curious if there was an analysis for the entire age range.
Response: In this study, we included all MABC strains isolated in our reference laboratory. As expected, as MABC infections are known to be associated with elderly patients, between 2014-2022 MABC cases detected in Portugal were mainly from these age groups.
3-2. It might be beneficial to list the data used in Figure 1 by year, unless there is a specific reason for the current arrangement.
Response: We thank the reviewer for this suggestion. We have kept the organization by subspecies but have ordered the isolates by year.
- Discussion:
Since the paper's structure separates Results and Discussion, it's advisable to provide links to specific data discussed in the Discussion section.
Response: This has now been done throughout the Discussion section.
- Consider adding a brief conclusion. This is essential for summarizing the paper effectively.
Response: We believe that the brief conclusion corresponds to the final paragraph of the Discussion section (Line 329-340). We leave to the editor the decision to separate this paragraph into a differently titled “Conclusions” Section.
Round 2
Reviewer 2 Report
The authors have successfully implemented my suggestions, and the manuscript has been significantly improved. Now, this paper is more than adequate for publication in IJMS. Congratulations!